# Free and Open Source Software organizations: A large-scale analysis of code, comments, and commits frequency

**Tadeusz Chełkowski, Dariusz Jemielniak**  **\*, Kacper Macikowski**

MINDS (Management in Networked and Digital Societies), Department at Kozminski University, Warszawa, Poland

\* darekj@kozminski.edu.pl

## Abstract

As Free and Open Source Software (FOSS) increases in importance and use by global corporations, understanding the dynamics of its communities becomes critical. This paper measures up to 21 years of activities in 1314 individual projects and 1.4 billion lines of code managed. After analyzing the FOSS activities on the projects and organizations level, such as commits frequency, source code lines, and code comments, we find that there is less activity now than there was a decade ago. Moreover, our results suggest a greater decrease in the activities in large and well-established FOSS organizations. Our findings indicate that as technologies and business strategies related to FOSS mature, the role of large formal FOSS organizations serving as intermediary between developers diminishes.

## Introduction

Online communities in general and Free and Open Source (FOSS) communities in particular, have been a subject of stable academic interest since their inception [1–3]. Although individual FOSS projects have been the subject of many in-depth analyses, the organizations that manage and control FOSS projects have not yet garnered much academic interest.

Within the field of organization studies, researchers have studied topics such as the emergence of new teams from FOSS development networks [4], continued engagement [5], successful productization of peer production in software [6], group activity, dynamics, and social ties [7,8], diversity [9], leadership [10,11], interdependencies [12], the influence of leaders on project sustainability [13], network ties between projects [14], or and IP strategies [15].

As a new phenomenon, FOSS has often been described in terms of its innovative nature, market potential [16], surprising growth [17], the ability to "hack" capitalism [18], and its key differences from traditional software [19]. The gist of much of the literature is that the peak of the FOSS revolution is ahead of us and that we are still observing its growth and maturation [20,21] as organizational and economic regimes continue to change [22].

It is worth noting that "free software" and "open source software" are similar, but not identical, especially in activist circles where they are hotly debated. In order to avoid complex, ideological, and licensing-nuanced discussions we therefore attempt to stay neutral and use Free/Open Source Software (FOSS) as a caveat term [23–25].

**Funding:** This study was funded by the Polish National Agency for Academic Exchange (NAWA) to DK (PPN/BEK/ 2018/1/00009). The funders had no role in study design, data collection and analysis, decision to publish, or preparation of the manuscript.

**Competing interests:** The authors have declared that no competing interests exist.

Using "FOSS" to refer to free and/or open source software is a way to capture two different philosophies: the one formulated by Richard Stallman in 1983 and Open Software as defined by Open Source Initiative [26]. We acknowledge that many researchers have traced the roots of open source to early as 1970 [27], but we understand that the term "open source" was coined in 1998 to separate the free programs from Open Source Initiative's ideas of freedom. The term "Open Source" captures two distinct ideas, therefore it's worth emphasizing that despite even though in many cases "Open Source" is used as a single term, it refers to two separate movements within the free software community. The first is the mission of promoting computers' freedom to use software without any cost and copyright restrictions. The second refers to a more practical aspect of making software source codes accessible [28,29]. FOSS now incorporates philosophies and approaches as distant as leftist activism and corporate strategies [27]. For our purposes we are going to refer to FOSS mainly in its politically neutral field of collaborative and organizational practices. See also: https://gnu.org/gnu/the-gnu-project.html and https://opensource.org/osd.

In the early 21$^{st}$ century it seemed that FOSS would revolutionize society. Wikipedia conquered the market for online encyclopedias and marginalized Britannica [30], Linux became the No.1 server, breaking Microsoft's monopoly [31], and Firefox was the most popular browser after Internet Explorer bundled with Windows [32]. All these successes led some researchers to hypothesize that FOSS in particular, and peer production in general had the potential to transform late capitalism [18,33]. Sharing and cooperation were expected to emerge as a new modality of economic production [34], leading to a groundbreaking transformation of markets and societies [35]. FOSS, through the creation of new forms of property, would "infect capitalism like a virus," and challenge the dominant logic of private property and ownership [36–38]. The emergence of private collectives [39], creating new, a-hierarchical and loosely coordinated structures [29], and relying on creation of zero-reproduction costs goods, often of a non-competitive character [36] offered the promise of an entirely new organizational model that would gradually take over the existing ones. They also indicated a fundamentally different approach to organizational innovation [40].

On the surface, the narrative about constant growth and increase in importance seems very plausible. The development and global diffusion of FOSS are quite clear [16]. Even though some projects are naturally abandoned [41], there are certain patterns of growth and decline in FOSS projects [42], and we can reasonably expect FOSS organizations to grow and take over an increasing portion of market share from traditional organizations [43–45].

In the early 1980s, the open source community grew and open source sharing customs were embraced by a growing number of academic and non-academic organizations. In 1985, as a result of the conflict between AT&T and UNIX, Richard Stallman created a Free Software Foundation protecting the right to keep software freely available [46]. The institutionalization of the open source movement produced a variety of organizations structured around an idea, a project, a group of projects, or more recently, software vendors [35,47]. Once an open source group of collaborators reaches a certain size, the norms of sharing, licensing standards and maintenance duties of the community need to be maintained. FOSS organizations adopt or design FOSS licensing standards, distribution methods, software development standards, outside world communication representatives, quality and testing procedures and finally tools for community collaboration. FOSS organizations adopt traditional controlling structures to a degree that was needed to control the release process, but at the same time relaxed enough to preserve the free nature of the open source movement [46–48].

However, even though digital commons, peer production, and open collaboration are still perceived as showing great promise [49,50], and the dream of open organizing's transformative powers has not been entirely lost [51], the situation has become much more fuzzy in the

past decade. While FOSS has always had some balance of for-profit and for-fun activities [26], large corporations have recently been able to incorporate elements of FOSS organization and approach into their traditional business development strategies [1], and to exploit FOSS software for closed and proprietary products [52]. In fact, even though the FOSS model initially proved a viable alternative to traditional software development methods, it has not been consistently successful in productization: the creation of products that the customers would find easy to understand and use [6,53]. Open organizing is a beautiful idea that showed enormous promise when it took the traditional modes of organizing by surprise, but it may be already past its peak. It is also much more hierarchical and bureaucratic than it originally assumed [54].

To understand the future and place of FOSS in management and society, it is more important than ever to measure engagement in FOSS projects over time across selected small, medium, and large projects. It should allow both the estimation of the general development of FOSS, and reveal the finer details, depending on the size of the organization. Our paper is an attempt to fill this gap.

## Project rationale

The Apache Software Foundation is often cited as a paragon of FOSS organization [55–57]. According to Mark Driver, research vice president at Gartner, "The Apache Software Foundation is a cornerstone of the modern open source software ecosystem–supporting some of the most widely used and important software solutions powering today's Internet economy." (https://blogs.apache.org/foundation/entry/apache-is-open). Indeed, the Apache Software Foundation (ASF) is arguably the most prominent example of a large and successful FOSS organization. It is responsible for the fundamental components of the modern web architecture (Apache HTTP) [58], the backbone of data mining (Apache Hadoop, Apache Spark) and hundreds of tools essential for programming, integration and standardization of the internet as we know it [16,59,60].

Since 1999, Apache has been not only a place for project development, but also a model of open innovation and open collaboration, in many cases displacing traditional software development methods. However, although the Apache Software Foundation is proud of its continuous growth, it is worthwhile to look more closely at the fine-grained details of the community's activity. For instance, data presented on the official Apache statistics pages (https://projects.apache.org/statistics.html). indicate an undeniable success in the growing code base; however, activity measured in community emails and issues presents very interesting fluctuations. According to Fig 1, ASF recorded the highest number of emails (78 846) in March 2016 but that number dropped to 42 814 in October 2017, a level last seen in May 2011 (42 400).

The observed decrease in the email communication can be explained by factors such as change in users' behavior: many users moved away from email to integrated messaging systems in the code repository interface [61,62]. At the same time, there is empirical evidence of the correlation between the ratio of email messages in public mailing lists to versioning system commits [63] and consequently to project activity as a whole. Thus, it may be a possible signal of decreased participation in FOSS projects. However, even though emails and communication in FOSS have been studied as a proxy for project health and growth [56,64], and mature projects are known to rely on well-structured communication [65], research so far has focused on small samples, precluding a more definitive observation of larger trends over time. This observation has inspired us to conduct what we believe is one of the largest analyses of FOSS projects' code, gathering data from 1314 individual projects and 1.4 billion lines of code managed.

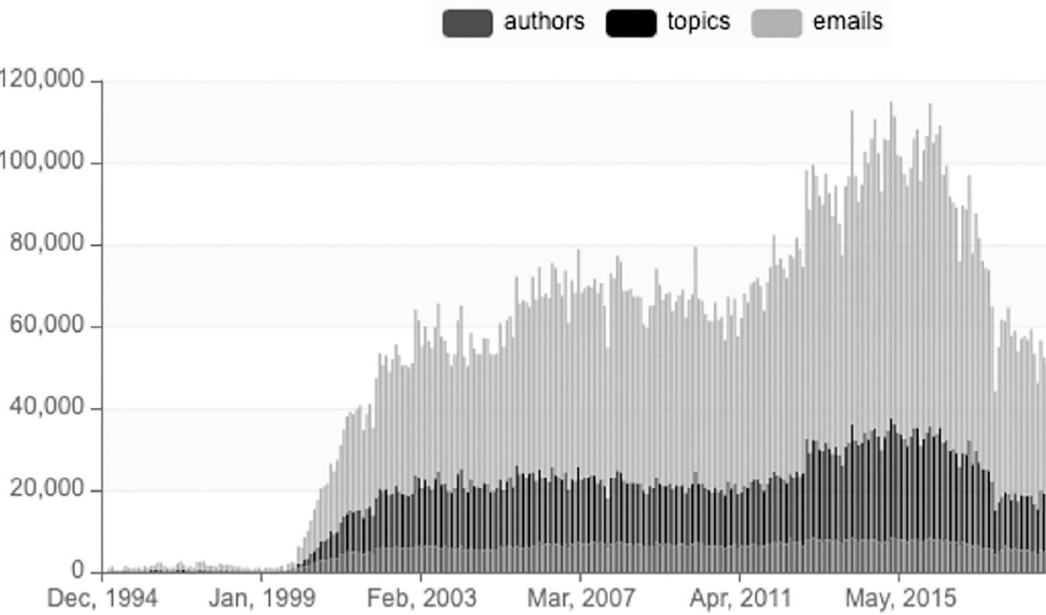

**Fig 1. Emails, topics and authors.** Source: https://projects.apache.org/statistics.html.

The strength of our study relies on a huge sample of commits, which allows us to make more certain observations about the changes, even though it also makes providing explanations more difficult. Additionally, one advantage of our study is its long-time focus: as only in the longer periods it is possible to observe incremental but clear shifts in the organizational landscape.

## Research question

The aim of our study is to improve our understanding of the level of activity among a large sample of FOSS projects. Additionally, data stratification on the FOSS organization gives us a chance to analyze projects from the perspective of the FOSS organizational association. The importance of FOSS organizations such as Apache Software Foundation for the modern networked society could not be overestimated; any change in its community dynamic is an interesting factor from the vantage points of academia and business [60,61]. In order to understand it better, we explore the following research question:

What is the structure of commits, code and comments contribution among the selected Open Source Software Organizations over the last 20 years?

For this article, we've selected a stratified sample of small, medium and large FOSS organizations. We have collected 21 years of quantitative data describing commits frequency, code inserts and deletions as well as data about comments attached to code.

We classify this study as exploratory research on a large data sample, a first step in the direction of deeper case-oriented analysis. In order to answer our research question, we've quantified contributors' activities on the project level. To picture the activity across the analyzed projects, we use simple contingency tables collecting commits, comments, code projects as our main variables and its relationship as calculated variables. We argue that activity in FOSS projects measured in commits, source code, and comments has declined over the last 10 years.

## Materials and methods

### Data source

Our research data such as number of commits, comments and code lines was collected from Open Hub, a database and public directory of FOSS (Open Hub and the Open Hub logo are trademarks of Black Duck Software, Inc. in the United States and/or other jurisdictions). Open-Hub's automated analytics software regularly visits the most popular source versioning systems such as Git, SubVersion, CVS and Bazaar. OpenHub curates data using its large online community where anyone is able to correct and edit OpenHub data entries. It is now arguably the largest and most trustworthy aggregated data source of FOSS. This article is based on the May 20, 2018 snapshot of the OpenHub.net repository. Since the frequency of updating the OpenHub repository may vary, APIs of the developers repositories change, and OpenHub needs time to adapt to these changes, to avoid data inconsistencies we have not used data after December 31, 2017, so as to make sure that the data completeness is as high as possible. Using the custom developed application and publicly available OpenHub API, we have collected a large data sample containing a comprehensive overview of the 20-year history of FOSS organizations.

### Data sample and data collection

The result of programmers' work are code lines and comment lines distributed among the number of files, in most cases compiled into an executable software. Programmers are generally producing software using programming languages that fall into one of two categories: interpreted or compiled. Interpreted programming language code must be parsed, and executed each time the program is run. The compiled programs are translated by compilers into a very efficient lower level code that can be executed many times. Some programming languages are using a dual interpreted or compiled paradigm. The main artifact created in the process is a source code represented as a set of statements written in a programming language like Java, C, C++, JavaScript or XML, CSS or HTML tags. Software developed in collaborative environments is created in a series of commits; commits happen every time a developer wants to contribute a piece of work to a shared repository. This process is supported by concurrent developments in software such as Git [66].

The role of concurrent software development applications is to track the changes between the programmer's local environment and synchronize it with a remote repository, making sure that potential code changes and code conflicts are resolved and seamlessly merged (see: https://git-scm.com/about). A source-code modification such as adding, modifying, or removing lines of code, adding or removing files, changes in the documentation files, are typical examples of commits. Because of the open nature of software repositories and their accessibility, commits have been a subject of numerous software development studies [67,68], and the activity of developers measured in commits is known to be highly unequal in FOSS organizations.

Additionally, to make a source code clearer and easier for others to understand, programmers add comments to address the meaning of the code block or a code line. As noted by researchers, "source code comments are a valuable instrument to preserve design decisions and to communicate the intent of the code to programmers and maintainers" [69].

For the purpose of our research, each month we analyze the number of code lines and comment lines added by programmers for each project. To retrieve and collect the research information, we have developed an automated application retrieving and parsing data using the headless interface of OpenHub. Our application, which relies on REST API, listed the requested organizations and records, reflecting committers' activity ordered by projects in monthly snapshots.

## Dataset selection and stratification

FOSS organizations differ in many ways–some like Eclipse represent large companies and their business goals while others like Apache others started as a single open project which, over time, attracted more programmers with new projects and ideas.

To reduce sampling error and improve the precision of the results, we've divided the FOSS population into homogeneous subgroups before sampling *(stratification)*. Subgroups *(strata)* were determined by the size of the FOSS organization, measured in numbers of projects [70–72]. Selection of a project as a stratification criterion has limitations that are discussed in the limitations, data and results section.

Stratum 1 - [LARGE] organizations with managed #Projects > = 100
Stratum 2 - [MEDIUM] organizations with 100> #Projects > = 25
Stratum 3 - [SMALL] organizations with #Projects <25

Combined sample consists of (n = 1314) projects with MOE (Margin of Error) ±2.30% for the CL (Confidence Level) = 95% and MOE ±3.03% for the CL = 99%). It encompasses 15 FOSS organizations, 16 727 184 commits and over 1.4 billion lines of code. The collected attributes timespan ranges from 11 to 21 years. For each project in each year, we have collected a full 12-month history or a partial history (some project life spans are shorter than 21 years). In total, we have collected 3246 data months, in 9 cases the year data did not include the all months data.

## Data record

Each record consists of raw attributes imported from data sources and variables derived from the collected data. Individual record represents a monthly activity for the analyzed project managed by the FOSS organizations. To understand the nature of projects' activity better, we have calculated additional attributes.

First, for better understanding of the project committers' level of activity and the nature of developed software, we measure a coefficient of code submitted per commit using the following equation:

CODPC = ΣLines of Code/ΣCommits

CODPC might indicate the current project stage as frequent commits with a low number of submitted code may indicate that a project is in the maintenance phase [55]. Second, to identify the relationship between the lines of comments submitted in a single commit, we calculate comments per commit coefficient.

COMPC = ΣComments/ΣCommits

This might be an interesting indicator of, for instance, the documentation phase of the project and code creation phase [55]. Lastly, we have calculated the ratio of comments per line of effective source code,

COMPCOD = ΣComments/ΣCode

since a high number of comments per line of actual code may indicate more formal organization processes in a project. The list of variables, with types and source is demonstrated in Table 1.

Sample data record is presented in Table 2.

The number of collected observations varies among organizations, which is well represented in the frequency table (see Table 3).

A comparison of projects, commits, and code size is included in Table 4.

## Results

### Commits' analysis

Fig 2 shows that the drop in the commits' volume growth affecting large FOSS organizations (stratum 1) started around 2010, and continued to the end of the dataset history. A closer look at the

**Table 1. The list of variables, with types and source classification.**

| Variable name | Description | Variable type | Source |
|---|---|---|---|
| Organization name | Open Source Software Organization name | Classification variable | Collected from the Data Source |
| Project name | Name of the Open source project | Classification variable | Collected from the Data Source |
| Organization size | Classification of the organization size—Large, Medium or Small. | Classification variable | Calculated, using managed projects number as a determination criterion. |
| Lines of code added | Number of code lines added in the observed time (one month). | Main quantitative variable | Collected from the Data Source |
| Lines of comments added | Number of comments lines added in the observed time (one month) | Main quantitative variable | Collected from the Data Source |
| Blank Lines added | Number of blank lines added in the observed time (one month) | Supporting quantitative variable | Collected from the Data Source |
| Commits | Number of commits lines added in the observed time (one month) | Main quantitative variable | Collected from the Data Source |
| Lines of code removed | Number of code lines removed from the code commit-to-commit comparison in the observed time (one month) | Supporting quantitative variable | Collected from the Data Source |
| Lines comments Removed | Number of comments lines removed from the code commit-to-commit comparison in the observed time (one month) | Supporting quantitative variable | Collected from the Data Source |
| Blank lines removed | Number of blank code lines removed from the code commit-to-commit comparison in the observed time (one month) | Supporting quantitative variable | Collected from the Data Source |
| Code per commits (CODPC) | Calculated value. Code/commits. Variable calculated for all collected cases. | Supporting variable. | Calculated variable. |
| Comments per commit (COMPC) | Calculated value. Comments/commits. Variable calculated for all collected cases. | Supporting variable. | Calculated variable. |
| Comments per code (COMPCOD) | Calculated value. Comments/code. Variable calculated for all collected cases. | Supporting variable. | Calculated variable. |
| Year | Time variable, observation year. | Main data variable | Collected from the Data Source |
| Month | Time variable, observation month. | Main data variable | Collected from the Data Source |

data with trimmed mean (top and bottom 5% of observations have been removed), reveals that the average annual growth for large FOSS organizations was 25.39%, 30.21% for medium FOSS organizations (stratum 2) and 35.68% for the small FOSS organizations (stratum 3). Table 5 presents the combined growth rates for FOSS organizations of all three studied organizations sizes.

**Table 2. Example of collected data record.**

| | |
|---|---|
| Organization | Apache |
| Size | Large |
| Project | Apache Commons Math |
| code added | 257 |
| code removed | 891 |
| comments added | 24 |
| comments removed | 449 |
| blanks added | 17 |
| blanks removed | 136 |
| Commits | 10 |
| contributors | 1 |
| Year | 2016 |
| Month | 12 |
| CODPC | 25.7 |
| COMPC | 2.4 |
| COMPCOD | 0.093385214 |

**Table 3. Frequency distribution of collected observations.**

|  | Frequency (*Fc*) | Percent | Cumulative Percent |
|---|---|---|---|
| Apache | 22762 | 26.5 | 26.5 |
| Debian | 3555 | 4.1 | 30.6 |
| Eclipse | 14666 | 17.0 | 47.6 |
| Gentoo | 1314 | 1.5 | 49.2 |
| GNOME | 245 | .3 | 49.5 |
| JBoss | 2821 | 3.3 | 52.7 |
| Kde | 18448 | 21.4 | 74.2 |
| Mozilla | 5866 | 6.8 | 81.0 |
| nasa | 1918 | 2.2 | 83.2 |
| openstack | 1091 | 1.3 | 84.5 |
| OSGeo | 3073 | 3.6 | 88.1 |
| OW2 | 2788 | 3.2 | 91.3 |
| OWASP | 1320 | 1.5 | 92.8 |
| tdf | 248 | .3 | 93.1 |
| wikimedia | 5909 | 6.9 | 100.0 |

## Code analysis

Fig 3 shows the source code growth dynamic, measured in the number of lines written. It is worth to notice differences between the source code contribution between the three analyzed groups and a dominance of the medium FOSS organizations. In order to have a clear view of the code base we use the code net value as a variable. Code net value represents a number of functional code lines, without banks and comment lines, additionally it deducts the deleted lines, since even a single commit can add and also remove code.

**Table 4. Projects, commits and code size comparison.**

|  | # of projects | Σ commits | Σ code added | Σ code removed |
|---|---|---|---|---|
| Large (#projects>100) |  |  |  |  |
| apache | 343 | 1 828 824 | 407 946 922 | 275 627 732 |
| eclipse | 172 | 1 532 770 | 449 599 501 | 295 623 314 |
| kde | 204 | 4 619 403 | 768 319 949 | 521 996 888 |
| nasa | 111 | 96 374 | 44 976 530 | 25 955 044 |
| wikimedia | 168 | 1 164 674 | 63 501 047 | 44 812 409 |
| Medium (25<#projects<100) |  |  |  |  |
| debian | 29 | 789 435 | 710 261 271 | 460 060 995 |
| JBoss | 35 | 583 720 | 164 351 117 | 117 092 382 |
| mozilla | 94 | 2 559 615 | 777 124 625 | 516 245 203 |
| OW2 | 40 | 501 288 | 519 017 687 | 260 478 326 |
| OWASP | 63 | 49 877 | 31 375 648 | 23 233 759 |
| Small (#projects< = 25) |  |  |  |  |
| gentoo | 14 | 768 767 | 39 103 075 | 28 916 805 |
| GNOME | 1 | 523 280 | 37 933 312 | 28 153 328 |
| openstack | 13 | 926 429 | 76 950 341 | 51 594 436 |
| OSGeo | 25 | 383 002 | 70 590 585 | 48 623 746 |
| tdf | 2 | 399 726 | 60 838 350 | 41 705 135 |
| Total | 1 314 | 16 727 184 | 4 221 889 960 | 2 740 119 502 |

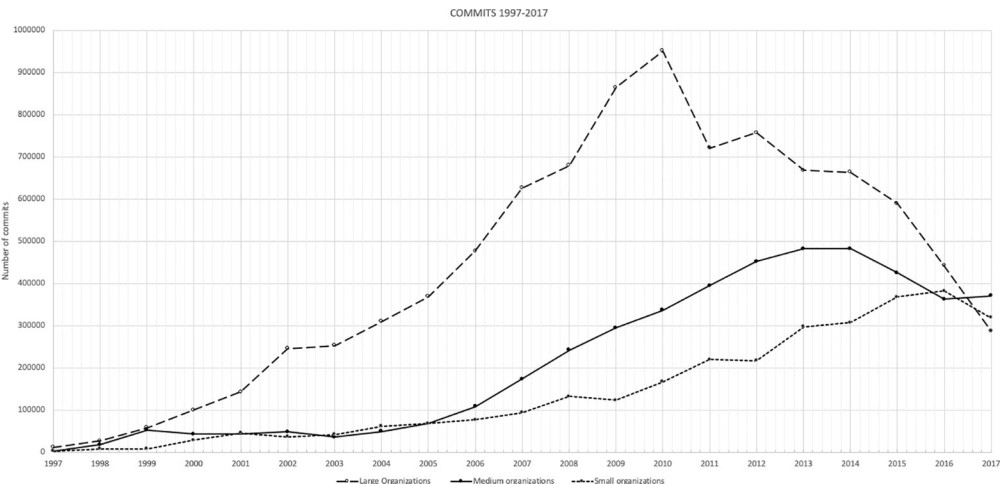

**Fig 2. Commits analysis in FOSS organizations 1997–2017.**

## Comments' analysis

In order to have a clear picture of comments contribution among the analyzed FOSS organizations we have introduced a new metric: contributed lines of comments per lines of code (COMPCOD). COMPCOD, described in Table 6, shows differences among large, medium and small FOSS organizations in their code commenting behavior. In the medium FOSS organizations, the largest average value of 0.86 lines of comments per code line was recorded. It's important to emphasize that within the collected observations, only one organization, The Open Web Application Security Project (OWASP), is an outlier with over 2.46 comment lines per code line. Tables 6–8 show the mean COMPCOD, mean of comments line per code lines, and the number of source code lines per commit.

## Discussion

Although all the analyzed organizations have grown over the past 20 years (Figs 2 and 3) we have observed lower and decreasing growth rates in the large FOSS organizations when compared to medium and small FOSS organizations (stratum 3). Furthermore, in recent years the commits volume of large FOSS organizations (stratum 1) started to drop by an average of 16.7% annually. The best example of this trend is the fact that in 2017, small FOSS organizations (stratum 3) surpassed the large FOSS organizations (stratum 1) commits volume by 10.8%. This is surprising, as 10 years earlier the commits volume of large FOSS organizations (stratum 1) was more than 6 times bigger than that of small FOSS organizations (stratum 3) (626 136 to 93 512) and over three and a half-time bigger than medium FOSS organizations (stratum 2) (626 136 to 172 843) (Fig 2).

**Table 5. Combined growth rates for large, medium and small FOSS organizations 1997–2017.**

|  | Large | Medium | Small |
|---|---|---|---|
| Trimmed mean (5%) | 25.59% | 30.21% | 35.68% |
| Mean | 28.27% | 56.17% | 44.42% |
| Max | 142.27% | 631.01% | 273.87% |
| Min | -34.88% | -25.42% | -18.95% |

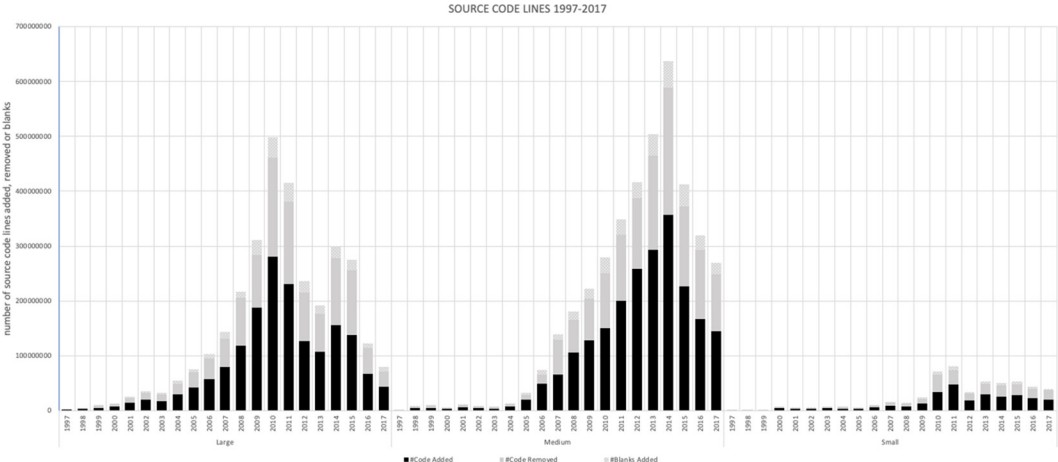

**Fig 3. Source code committed in number of lines.**

Moreover, as Fig 2 shows, the drops in the commits' volume growth is affecting large FOSS organizations disproportionately, and this phenomenon was not observable in the first 10 + years. As Table 5 shows, the trimmed mean (top and bottom 5% removed) the change is dramatic: The drop in the large FOSS organizations activity measured in commits can be demonstrated by one comparison—commits in 2017 represented only approximately 30% of commits compared to the record high in 2010 (2010–951 294, 2017–287 907).

In the first decade of the period under analysis (1997–2017), we observed a steady growth in the medium and large FOSS organizations, while small FOSS code growth dynamics tended to fluctuate. However, from 2010, the organizations managing over 100 projects started to receive less new source code than in the previous decade. Compared to 2010, when users

**Table 6. Mean COMPCOD (mean).**

|  | Mean of comments lines per code line |
| --- | --- |
| Large(#projects>100) |  |
| apache | 0.61270 |
| eclipse | 0.51178 |
| kde | 0.32221 |
| nasa | 0.39455 |
| wikimedia | 0.32492 |
| Medium(25<#projects<100) |  |
| debian | 0.30002 |
| JBoss | 0.43842 |
| mozilla | 0.44154 |
| OW2 | 0.68189 |
| OWASP | 2.48573 |
| Small(#projects< = 25) |  |
| gentoo | 0.22984 |
| GNOME | 0.14378 |
| openstack | 0.20929 |
| OSGeo | 0.44055 |
| Tdf | 0.67851 |

**Table 7. Mean of comments line per code lines (COMPCOD).**

| Mean values of comments lines per code line (COMPCOD) | Large(#projects>100) | Medium (25<#projects <100) | Small (#projects< = 25) |
|---|---|---|---|
| 1997 | 0.15529 | 0.25613 | 0.16807 |
| 1998 | 0.14792 | 0.24806 | 0.21714 |
| 1999 | 0.21548 | 1.15850 | 0.31579 |
| 2000 | 0.26096 | 0.26674 | 0.28325 |
| 2001 | 0.31828 | 0.60631 | 0.23779 |
| 2002 | 0.28140 | 0.29048 | 0.33766 |
| 2003 | 0.47377 | 0.93073 | 0.65974 |
| 2004 | 1.36562 | 1.60681 | 0.37094 |
| 2005 | 0.42731 | 0.43290 | 0.36057 |
| 2006 | 1.07082 | 0.31924 | 0.31417 |
| 2007 | 0.37597 | 1.00306 | 0.27286 |
| 2008 | 0.63819 | 0.29977 | 0.26000 |
| 2009 | 0.36495 | 0.32552 | 0.28688 |
| 2010 | 0.64023 | 2.14643 | 0.95249 |
| 2011 | 0.35907 | 0.26990 | 0.23549 |
| 2012 | 0.44546 | 0.34084 | 0.49329 |
| 2013 | 0.33473 | 0.45488 | 0.24373 |
| 2014 | 0.31285 | 0.50129 | 0.22618 |
| 2015 | 0.29581 | 0.89144 | 0.33060 |
| 2016 | 0.41132 | 0.23855 | 0.32675 |
| 2017 | 0.59586 | 0.21995 | 0.22574 |
| Trimmed mean (5% bottom and 5% top) | 0.4199 | 0.5495 | 0.3157 |
| Mean | 0.4520 | 0.6099 | 0.3390 |
| SD | 0.27530 | 0.50011 | 0.16956 |

committed over 100 million of the net new source code lines, in the year 2012 large FOSS organizations (stratum 1) received only 36.1 million lines. In that same period, medium FOSS organizations (stratum 2) surpassed large FOSS organizations and by 2012 they had received almost 3.5 times more net new source code than large FOSS organizations. One of the most surprising findings is a noticeable "time shift" of the growth dynamic between the medium and large FOSS organizations. The observed decrease in the code base growth dynamic starts two years later in medium FOSS organizations (stratum 2) and two years later than that in small FOSS organizations.

In a deeper analytical look at our proposed metric COMPCOD (Table 6), studying the distribution reveals that one of the OWASP projects, "The OWASP Zed Attack Proxy," described as ". . . one of the world's most popular free security tools and is actively maintained by hundreds of international volunteers," (https://github.com/zaproxy/zaproxy/wiki) includes the code of conduct, instructions and even the elements of documentation in the comments sections. Regardless of the outliers, projects associated with large FOSS organizations registered less activity than projects in the medium FOSS organizations (stratum 2) with comments per line 0.45 ratio.

Finally, small FOSS organizations (stratum 3) are the least active in code comments, providing approximately 1 line of comment for every 3 lines of code (COMPCOD = 0.34). Additionally, an analysis of 20 years of comments per code history shows that standard deviation in large and small FOSS organizations (stratum 3) is smaller.

**Table 8. Number of source code lines per commit (CODPC).**

|  | Large(#projects>100) | Medium (25< #projects<100) | Small (#projects < = 25) |
|---|---|---|---|
| 1997 | 159.6457 | 213.6047 | 188.9899 |
| 1998 | 121.7883 | 1367.6196 | 63.7804 |
| 1999 | 122.4032 | 99.2152 | 585.9229 |
| 2000 | 100.4900 | 95.5775 | 169.4701 |
| 2001 | 137.2856 | 129.0426 | 116.8105 |
| 2002 | 112.5782 | 169.6384 | 147.2365 |
| 2003 | 75.3101 | 217.3441 | 133.9808 |
| 2004 | 202.8971 | 304.0662 | 77.2385 |
| 2005 | 254.2520 | 251.1138 | 107.8710 |
| 2006 | 208.5257 | 484.9788 | 188.4939 |
| 2007 | 209.4148 | 373.9491 | 165.4688 |
| 2008 | 218.2877 | 374.0096 | 110.8150 |
| 2009 | 212.1694 | 300.7204 | 149.9105 |
| 2010 | 222.8003 | 296.1230 | 206.8572 |
| 2011 | 252.7280 | 376.7635 | 352.3639 |
| 2012 | 218.6569 | 715.4053 | 131.8826 |
| 2013 | 209.0371 | 500.2177 | 131.1219 |
| 2014 | 226.6461 | 612.1060 | 130.7347 |
| 2015 | 250.4403 | 1202.5672 | 102.1072 |
| 2016 | 215.5444 | 555.9114 | 105.3622 |
| 2017 | 227.4132 | 184.8912 | 143.2757 |
| Trimmed mean (5% bottom and 5% top) | 190.9870 | 387.4562 | 150.5258 |
| Average | 188.4912 | 420.2317 | 167.1283 |
| SD | 51.88962 | 251.80445 | 110.31318 |

## Conclusions

Our results indicate a shift in contribution activity across FOSS projects of different sizes and growth stages over time, and that the largest organizations are slowing their growth at a faster pace than medium and small organizations.

There are many possible reasons for the observed phenomenon. We cannot exclude the possibility that the modalities of cooperation have changed over time and that the measures we are using do not hold a stable accuracy over the whole period.

However, if the results reflect the actual changes in FOSS organizations and projects, they are quite troubling for the open source movement.

One possible interpretation of this phenomenon is that open source, as an approach to developing projects, has lost some of its appeal. It is worth remembering that at first, FOSS principles were interpreted as bringing together an ideological paradigm shift (openness), governance and technological innovations [52,73]. These three areas were conflated into one, and raised the hopes of early enthusiasts that openness as a social norm is inseparable from and consequent to the other two, and supports a redefinition of labor leading to the reshaping of capitalism. In other words, the dominant assumption was that as new forms of governance and technology promote open organizing, we can expect traditional organizing to be gradually replaced, and the far-reaching consequences, according to some authors, may even change the capitalism as we know it [35].

Our study does not allow us to make claims about causality, and as our interpretation here is speculative, it should be treated with caution. However, what we believe may be happening

is the result of FOSS technology and organizational model maturing and becoming mainstream. While initially the governance and technological innovations indeed led to a wider adaptation of openness as a dominant logic, the traditional organizations soon learned how to use (and sometimes abuse) these two innovations to create closed ecosystems and gatekeep their position.

The successes of Google in leveraging Android to win commercially on a mobile market, or of WordPress to build a regular business based on FOSS principles, as well as a series of take-overs, such as acquiring GitHub by Microsoft for 7.5 billion dollars, and acquiring RedHat by IBM for the staggering sum of 34 billion dollars, all show that rather than transforming society, FOSS may be trimmed and harnessed for traditional corporate goals. While open source may be on the rise as an effective organizing principle [74], it has been disentangled from at least some of its original premises. The principles of sharing economy, rooted in collaborative, pro-social, and anti-commercial ideals [75] have also been used rhetorically and adjusted for the mainstream economy, leading to further exploitation and inequality [76]. In a way, FOSS movement has both "won and lost the war" [77], as it has been widely accepted as a form of software development, but the profits deriving from it have largely been appropriated by corporations. In its 2.0 version, FOSS development becomes yet another business model [78], bordering freemium more than a revolutionary society-changing movement.

The ideologies of openness, sharing, and collaborating are being repurposed for business as usual [79,80]. The openness of software have become routine factors for influencing productivity and efficiency [81,82]. Moreover, open collaboration software development turned out to be much less collaborative in an actual daily practice had been assumed [83,84].

Moreover, far from being stable, FOSS organizations underwent major adaptations to the environment. One of the major roles of FOSS organizations to nurture interactions among community members, calling actions, setting guiding principles or developing tools to facilitate collaborative software development and streamline coordination [85,86]. Benefits provided for the FOSS developers and users by the FOSS organizations, such as Apache Software Foundation, Mozilla Foundation or Linux Foundation, include project governance and vital institutional support infrastructure [87,88]. Users or contributors can rely on an organizational framework for intellectual property rights management as well as for legal support and well-defined development and maintenance processes. In many cases FOSS organizations exist as communities of practice, where people engage in collective development, learning and solving similar problems [89].

Yet, as technologies develop and organizational practices mature, some functions that had previously been crucial in FOSS development and provided by FOSS organizations may be replaced by software and online services. While this paper does not analyze the new emerging FOSS organizations especially created after 2017, the analyzed data provides evidence that activities measured as commits are declining, and may have dire side effects for the entire FOSS movement. The existence of large FOSS organizations has made big policy and activism possible. Promoting big ideological changes in the areas of open licensing, fairness in digital files sharing [90], sharing rather than selling as a principle of contemporary society, or openness in general as a strong social norm [91] would not have been possible without their support. Large FOSS organizations brought grand projects, such as new operating systems (Linux) or productivity suites (such as OpenOffice) into existence. These large projects were essential for the belief that the emerging peer-to-peer economy and the new commons may make a larger impact on the society that went beyond isolated cases of software [43,92].

It is possible that ideological manifestos, postulating openness as a new principle of social organizing, having a potential for transformative influence on capitalism, may not have had as much appeal as it seemed. Yes, FOSS organizations paved the way to distributed structures and to making openness an organizing principle, and according to some measures their

influence on capitalism may have been profound. They also developed tools and processes that made virtual collaboration more effective. Yet, our results may indicate that as soon as the traditional organizations caught up on both of these fronts, FOSS organizations, and especially the large ones, started to lose momentum.

It may be that the demand for a revolution simply was not there, and the general public couldn't care less about openness. Even though projects with a non-market sponsor, as well as with open licenses used to be able to attract greater user interest over time in the past [93], the successes of services such as TripAdvisor, Quora, Google Guides or Yelp have made it abundantly clear that many users do not have a problem with creating collective content for a for-profit company, which uses this content on a restrictive license, and relies on corporate-decided community governance without any open collaboration in regards to organizational structures and roles. They just enjoy a friendly UI, and a peer production mode of contributing. The final nail in the coffin has been the rise of centralized cloud services such as GitHub or BitBucket, which have met many of the organizational and cooperative needs of developers that were previously addressed by open designs.

## Limitations of the research model, data and results

Our study relies on data from the period of 1997–2017, and does not cover the most recent changes in the open source environment. While this approach is reasonable because of the data availability and comparability, it should be noted that in recent years FOSS organizations have explored new ways of supporting open source projects, and new ways of managing coordination, including ways more difficult to measure and compare to previous years.

This paper is a quantitative conceptualization of the activity levels in a stratified sample of projects associated with FOSS organizations. Even though OpenHub is a reliable source of data, the results should be considered within the trust boundary of the data source. There is no guarantee that all projects, commits, comments, or organizations are fully represented in the OpenHub database. It is also worth mentioning that the results are applicable only to FOSS projects associated with formal FOSS organizations, thus the results do not represent the full population of FOSS projects. The proposed perspective of looking at the FOSS organizations through the lens of projects, commits, submitted code, and code commits may not fully represent all behind-the-scenes activities, including important cooperative behaviors not related to coding, but providing the much-needed social glue of interactions. It is widely accepted that communication among FOSS collaborators happens in many different channels [94–97], and we have studied only the structured, technical ones. There are many activities that foster cooperation, and that are not code-centric [98,99]. Researchers used different methods to understand the nature of FOSS collaborations such as Social Network Analysis or dedicated metrics for understanding the nature of the FOSS models [58,63,84,100].

Despite these trends, our findings need to be reconfirmed through other methods. Our research raises many questions about the potential change in the way that FOSS processes are organized. Since the selected data source and perspective criteria introduce natural bias into our results, these results should not be unreflexively used to generalize to other FOSS communities or organizations. Moreover, as our results' main strength is the sample size, it is also its major weakness, as it makes an explanatory approach—seeking correlations, reasons, and causes–much more difficult.

## Final remarks

Our study is an attempt to determine the basic quantitative indicators of growth of FOSS organizations. We have discovered interesting trends in commits, comments and code growth

dynamics, indicating that there has been a change in the activity levels across all types of FOSS organizations. FOSS organizations are still gaining new code, but the collaborative efforts measured in commits, committed code, and comments are lower than they were in 2020. Medium and small FOSS organizations seem to be less affected by the overall slowdown, still attracting new users but not as quickly as in the past. These results might be explained by the increasing adoption of FOSS collaborative online services such as GitHub and BitBucket. With more tools and simpler collaborative processes there may be a diminishing need for organizational proxies, because people can create ad hoc short-lived structures without dedicated processes and formal committees. If the original success of FOSS was even partly a result of this form of organization substituting for what can be more easily achievable through online services and software tools, it is quite understandable that FOSS organizations develop less dynamically. However, if this is what is happening, the practical implications are considerable: instead of revolutionizing the society or even just software development, FOSS will turn out to be a modest innovation, one that temporarily helped resolve some structural and communication issues, but only until the mainstream organizations have absorbed some of its model, and until regular project management tools have sufficiently evolved.

Another possible explanation may be that we observe the maturation and aging of the FOSS development model: it not only no longer relies on archetypal hacking-for-fun, but it also has entered a stage in which many projects require maintenance and stability, and are much less reliant on frequent communication and commits. If this is the case, the FOSS model is not going to disappear any time soon, but it is still not going to make any radical organizational difference, and will remain a temporary fad in the organization of work.

Finally, we cannot exclude the possibility that the larger FOSS organization are all falling prey to the "rise and decline" phenomenon observed in Wikipedia and some other peer production projects [101,102], and rooted in the fossilization of procedures, and the growth of quality control systems.

We believe that in the near future we may observe a steady decline in the role of the large and formal organizations as large independent FOSS organizations are replaced by corporate-driven FOSS foundations. Perhaps the free software-oriented movement will reorganize itself into smaller, dynamic, tools-oriented networks. FOSS will probably not die, but it may not really live.

## Supporting information

**S1 Dataset.**
(XLSX)

## Author Contributions

**Conceptualization:** Tadeusz Chełkowski.

**Data curation:** Tadeusz Chełkowski, Kacper Macikowski.

**Formal analysis:** Tadeusz Chełkowski, Kacper Macikowski.

**Investigation:** Tadeusz Chełkowski, Dariusz Jemielniak.

**Methodology:** Tadeusz Chełkowski.

**Software:** Kacper Macikowski.

**Writing – original draft:** Tadeusz Chełkowski, Dariusz Jemielniak.

**Writing – review & editing:** Tadeusz Chełkowski, Dariusz Jemielniak.

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
