## [Decision Letter · Decision Letter 0]

28 Jan 2021

PONE-D-20-37277

Free and Open Source Software a large scale analysis of code, comments, and commit frequency

PLOS ONE

Dear Dr. Jemielniak,

Thank you for submitting your manuscript to PLOS ONE. After careful consideration, we feel that it has merit but does not fully meet PLOS ONE’s publication criteria as it currently stands. Therefore, we invite you to submit a revised version of the manuscript that addresses the points raised during the review process.

As you can see below, both reviewers identified interesting aspects of your manuscript. However they also raised a number of concerns that prevent it from meeting PLOS publication criteria. In particular, when revision your article, pay special attention to the following issues:

- Discussion of the obtained results: Both reviewers asked for a revision of the conclusions reached (and the way they were expressed), considering teh limitations of the study.

- Methodology: Notice specially the requirements by Reviewer 1 for a more clear description of the applied methodology. Moreover, address the concerns by reviewer 2 on the collected dataset.

- English: Reviewer 1 also asked for a revision of the English used, as some parts are much less clear than others.

We look forward to receiving your revised manuscript.

Kind regards,

Sergi Lozano

Academic Editor

PLOS ONE

Journal Requirements:

"no"

"no"

6. Please ensure that you refer to Figures 1, 2, 3, and 4 in your text as, if accepted, production will need this reference to link the reader to the figure.

7. Please include a copy of Tables 7, 8, and 9 which you refer to in your text on page 18.

Reviewers' comments:

Reviewer's Responses to Questions

**Comments to the Author**

1. Is the manuscript technically sound, and do the data support the conclusions?

Reviewer #1: Partly

Reviewer #2: Yes

2. Has the statistical analysis been performed appropriately and rigorously? 

Reviewer #1: I Don't Know

Reviewer #2: Yes

3. Have the authors made all data underlying the findings in their manuscript fully available?

Reviewer #1: Yes

Reviewer #2: No

4. Is the manuscript presented in an intelligible fashion and written in standard English?

Reviewer #1: No

Reviewer #2: Yes

5. Review Comments to the Author

Reviewer #1: My review is uploaded as an attachment. This following text is included to meet the minimum character count required by the PLOS ONE web form.

Lorem ipsum dolor sit amet, consectetur adipiscing elit, sed do eiusmod tempor incididunt ut labore et dolore magna aliqua. Ut enim ad minim veniam, quis nostrud exercitation ullamco laboris nisi ut aliquip ex ea commodo consequat. Duis aute irure dolor in reprehenderit in voluptate velit esse cillum dolore eu fugiat nulla pariatur. Excepteur sint occaecat cupidatat non proident, sunt in culpa qui officia deserunt mollit anim id est laborum.

Reviewer #2: SUMMARY

The paper presents a large study of FOSS projects to better understand the growth of FOSS organization via basic quantitative indicators. The dataset built for the analysis includes projects being developed by 15 FOSS organizations, which are classified into 3 groups (i.e., big, medium, small) according to the number of projects they manage. The identified groups are used to conduct the study. Results reveal that collaborative efforts (measured in terms of commits, committed code, and comments) are lower than at the beginning of the current decade. Only projects from medium and small FOSS organizations seems to be less affected by this trend.

REVIEW

The paper addresses an interested topic of the field and helps to better understand the evolution of the contribution activity in FOSS projects, in particular, in the context of FOSS organizations. Although the analysis is purely quantitative, its large size helps to visualize how collaborative actions (i.e., commits and comments) have changed along the years. This is revealing and worth tracking.

I have two main concerns:

* The collection date to build the dataset looks old, and it should be better noted in the "Limitations of the research model, data and results" section. My main concern is the exponential explosion of activity in collaborative development platforms such as GitHub, which are becoming the de facto standard to develop FOSS projects, and where the organization dimension sometimes is blurry. This is also related to possible novel ways to manage FOSS projects which may be beyond traditional FOSS organizations. I understand that this idea may require additional research, but the text should note the situation.

* The discussion and conclusions of the paper are mainly hypotheses about FOSS organization evolution, as no empirical confirmation (i.e., interviews with developers) has been done. I wonder if they could be presented first for FOSS projects in general, and then extrapolate for FOSS organizations (being very cautious). This concern is also related to the previous one, as in the last years (2017-2021) FOSS organizations have been actively working on exploring new ways to support and help FOSS projects. 

All in all, I believe the paper makes an interesting contribution, but the conclusions are too focused on FOSS organizations and such a link is hard to visualize from the data gathered and analyzed.

DETAILED REVIEW

The first section introduces the problem and presents most of the related work. It also presents the main triggering idea to conduct the analysis presented in the paper.

In the second section, the authors briefly discuss the differences between free and open-source software, which helps to present some characteristics (and issues) typically found in FOSS software. This also serves to motivate the study conducted in the paper.

The third section presents the research methodology of the paper. Some indications for the subsections:

* Table 1 could be simplified indicating the starting year and its collected data-months. When needed, years with less than 12 collected data-months should be indicated (e.g., openstack in 2017)

* When describing CODPC, COMPC and COMPCOD and referring to either (or comments), is that the difference between the added and removed code (or comments)?

The fourth section presents the results of the study. Companion tables and figures illustrated the obtained results.

The fifth and last sections discuss the results and elaborates on some hypothesis to justify them.

MINOR ERRORS AND POSSIBLE TYPOS

- Normalize the use of "open-source" vs "open source"

6. PLOS authors have the option to publish the peer review history of their article (what does this mean?). If published, this will include your full peer review and any attached files.

Reviewer #1: No

Reviewer #2: No

---

## [Decision Letter · Decision Letter 1]

20 Apr 2021

PONE-D-20-37277R1

Free and Open Source Software: A large-scale analysis of code, comments, and commit frequency

PLOS ONE

Dear Dr. Jemielniak,

Thank you for submitting your manuscript to PLOS ONE. After careful consideration, we feel that it has merit but does not fully meet PLOS ONE’s publication criteria as it currently stands. Therefore, we invite you to submit a revised version of the manuscript that addresses the points raised during the review process.

When revising your text, pay special attention to the following issues:

1.- Both reviewers pointed out that some aspects of the methodology are not properly justified and/or described. Notice PLOS ONE's publication criterion #3 (https://journals.plos.org/plosone/s/criteria-for-publication#loc-3).

2.- Reviewer 3 stressed the need for a better alignment between the results obtained and the conclusions reached. Notice also the journal's publication criterion #4 (https://journals.plos.org/plosone/s/criteria-for-publication#loc-4).

Moreover, before submitting your revised manuscript, please be sure that the comments by both reviewers are properly answered.

We look forward to receiving your revised manuscript.

Kind regards,

Sergi Lozano

Academic Editor

PLOS ONE

Reviewers' comments:

Reviewer's Responses to Questions

**Comments to the Author**

1. If the authors have adequately addressed your comments raised in a previous round of review and you feel that this manuscript is now acceptable for publication, you may indicate that here to bypass the “Comments to the Author” section, enter your conflict of interest statement in the “Confidential to Editor” section, and submit your "Accept" recommendation.

Reviewer #2: (No Response)

Reviewer #3: (No Response)

2. Is the manuscript technically sound, and do the data support the conclusions?

Reviewer #2: Yes

Reviewer #3: No

3. Has the statistical analysis been performed appropriately and rigorously? 

Reviewer #2: Yes

Reviewer #3: No

4. Have the authors made all data underlying the findings in their manuscript fully available?

Reviewer #2: Yes

Reviewer #3: Yes

5. Is the manuscript presented in an intelligible fashion and written in standard English?

Reviewer #2: Yes

Reviewer #3: Yes

6. Review Comments to the Author

Reviewer #2: I have not found an answer to my comments in the revised version of the paper. I hold my decision and attach again my concerns.

The paper addresses an interesting topic of the field and helps to better understand the evolution of the contribution activity in FOSS projects, in particular, in the context of FOSS organizations. Although the analysis is purely quantitative, its large size helps to visualize how collaborative actions (i.e., commits and comments) have changed along the years. This is revealing and worth tracking.

I have two main concerns:

* The collection date to build the dataset looks old, and it should be better noted in the "Limitations of the research model, data and results" section. My main concern is the exponential explosion of activity in collaborative development platforms such as GitHub, which are becoming the de facto standard to develop FOSS projects, and where the organization dimension sometimes is blurry. This is also related to possible novel ways to manage FOSS projects which may be beyond traditional FOSS organizations. I understand that this idea may required additional research which is not included in the paper, but the text should note the situation.

* The discussion and conclusions of the paper are mainly hypotheses about FOSS organization evolution, as no empirical confirmation (or attempt to confirm) has been done. I wonder if they could be presented for FOSS projects in general, and then for FOSS organizations (being cautious). This concern is also related to the previous one, as in the last years (2017-2021) FOSS organizations have been actively working on exploring new ways to support and help FOSS projects.

All in all, I believe the paper makes an interesting contribution, but the conclusions are too focused on FOSS organizations and such a link is hard to visualize from the data gathered and analyzed.

DETAILED REVIEW

The first section introduces the problem and presents most of the related work. It also presents the main triggering idea to conduct the analysis presented in the paper.

In the second section, the authors briefly discuss the differences between free and open source software, which helps to present some characteristics (and issues) typically found in FOSS software. This also serves to motivate the study conducted in the paper.

The third section presents the research methodology of the paper. Some indications for the subsections:

* Table 1 could be simplified indicating the starting year and its collected data-months. When needed, years with less than 12 collected data-months should be indicated (e.g., GNOME in 2017)

* When describing CODPC, COMPC and COMPCOD and referring to either (or comments), is that the difference between the added and removed code (or comments)?

The fourth section presents the results of the study. Companion tables and figures illustrated the obtained results.

The fifth and last sections discuss the results and elaborates on some hypothesis to justify them.

MINOR ERRORS AND POSSIBLE TYPOS

- Normalize the use of "open-source" vs "open source"

Reviewer #3: My review exceeds the length limit and I am uploading the full review as an attachment. A summary of my review is below:

This article undertakes a longitudinal study of participation in FOSS projects, considering commit frequency, lines of code, and code comments. The work concludes that contributors are less active now, and that this decline is more strongly associated with large organizations than smaller ones. This finding is associated with three outcomes: the relationship between peer production and commercial production is maturing, the role of large organizations as intermediaries is declining, and the "likelihood of new big formal organizations" managing "emerging" projects is "systematically declining". The work concludes that this result suggests that open organizing is a "pipe dream".

This paper tackles an important concern with an ambitious scope. The writing is clear, and I appreciate the attention to a range of FOSS-related organizations. The work has strong potential to contribute to the field, with implications for computer science, social computing, and organizational studies.To the extent these organizations are in a state of failure, we need to know about it.

Overall, however, I find that this project faces a crossroads; there is a disconnection between the evidence developed and the argument made, and one side or the other needs to give way. In particular, there needs to be a clear connection between all of the core elements of the paper. To assist in tracking my argument in parallel with the paper's argument, I have broken my concerns into six themes below: the unit of analysis, the sample, the measures, the methods, the connection between the analysis and the discussion, and the ultimate conclusions. However, the gist of my concern (inconsistency between the design section, analysis section, and introduction/discussion/conclusions) is threaded throughout the paper, so some of my concerns as I describe them may seem to repeat/reprise previous concerns; for this I apologize in advance.

7. PLOS authors have the option to publish the peer review history of their article (what does this mean?). If published, this will include your full peer review and any attached files.

Reviewer #2: No

Reviewer #3: No

---

## [Author Response · Author response to Decision Letter 1]

4 Jun 2021

Our full response is included as a separate file.

---

## [Decision Letter · Decision Letter 2]

30 Jun 2021

PONE-D-20-37277R2

Free and Open Source Software Organizations: A large-scale analysis of code, comments, and commit frequency using OpenHub.net data source

PLOS ONE

Dear Dr. Jemielniak,

Thank you for submitting your manuscript to PLOS ONE. After careful consideration, we feel that it has merit but does not fully meet PLOS ONE’s publication criteria as it currently stands. Therefore, we invite you to submit a revised version of the manuscript that addresses the points raised during the review process.

As you can see below, both reviewers are highly positive concerning your article. Nevertheless, Reviewer 3 provided a detailed list of issues (already included in a previous report) that should be addressed and/or properly discussed. In particular, your revision should pay special attention to point 2 in the reviewer's report (Evidence-Claims Link) as it is directly related to PLOS ONE's publication criterion #4 (https://journals.plos.org/plosone/s/criteria-for-publication#loc-4).

We look forward to receiving your revised manuscript.

Kind regards,

Sergi Lozano

Academic Editor

PLOS ONE

Journal Requirements:

Reviewers' comments:

Reviewer's Responses to Questions

**Comments to the Author**

1. If the authors have adequately addressed your comments raised in a previous round of review and you feel that this manuscript is now acceptable for publication, you may indicate that here to bypass the “Comments to the Author” section, enter your conflict of interest statement in the “Confidential to Editor” section, and submit your "Accept" recommendation.

Reviewer #2: All comments have been addressed

Reviewer #3: (No Response)

2. Is the manuscript technically sound, and do the data support the conclusions?

Reviewer #2: Yes

Reviewer #3: Partly

3. Has the statistical analysis been performed appropriately and rigorously? 

Reviewer #2: Yes

Reviewer #3: Yes

4. Have the authors made all data underlying the findings in their manuscript fully available?

Reviewer #2: Yes

Reviewer #3: Yes

5. Is the manuscript presented in an intelligible fashion and written in standard English?

Reviewer #2: Yes

Reviewer #3: Yes

6. Review Comments to the Author

Reviewer #2: I would like to thank authors for the detailed response and for addressing my comments. In my opinion, the new version of the paper has improved with regard to the previous one. I therefore recommend its acceptance.

Reviewer #3: I found myself even more excited about this paper having reviewed the latest version and I appreciate the revision efforts and response to comments. This is valuable and informative work that I am eager to see succeed. However, some of my core concerns remain, particularly with respect to the claims made versus the evidence presented.

Content-related concerns:

1. Abstract. I reprise my earlier comments here -- the final claim of the abstract is unsustained and needs to be amended. With apologies for the format of the below:

A. "as...the role of large FOSS organizations serving as platforms diminishes" -- it's not clear what is meant by "platforms" here, or whether these organizations seek to serve as such, or whether this service is indeed declining. What's presented is a decline in productivity and growth by various metrics, not a decline in "role as platform".

B. "the likelihood of new formal organizations" -- the analysis presented in this paper explicitly does not address new organizations nor does it measure likelihoods.

C. "managing the...projects emerging is systematically declining" -- the analysis does not address emerging projects; to claim a systematically declining likelihood, one would need evidence addressing the system rather than only its long-lived members (to capture 'systematically'), as well as measuring which organizations an emerging project goes on to join (to capture 'likelihood').

2. Evidence-Claims Link. The "organizations supplanted by platforms" argument is problematic and emblematic of the necessity to connect claims and evidence I cited in my previous review. The use of the term "platform" is unclear and the argument of platforms supplanting organizations is not sustained by the analysis presented.

A. In the abstract, organizations are said to have a role to serve as platforms; 1A has the opener for my points on this. Organizations create and use tools like GitHub to manage their work, but Mozilla for example is not Git nor does it frame itself as a Git provider.

B. The paper presents an argument that "platforms" are supplanting "organizations" (p. 16). This claim needs evidence. It may be the case that new project founders don't see a need to join an organization when a platform is sufficiently robust, but evidence for this is not presented because of the way the sample is constructed. Valid evidence here would perhaps look like joining rates of new projects through processes like the Apache Incubator versus similarly-successful projects staying independent. But in this case, there's still no supplanting per se because this is a false dichotomy: every project needs both an organization of some kind and an instance on github or similar, with sufficient value in terms of users and code already accumulated before they could hope to join any organization.

3. Sample. The response to my comments with respect to the sample are very much appreciated; a few more of these details should make their way into the paper, given the effort that was expended in this regard. I don't think it's necessary to include the data source in the title---my concern here is more about substance: reporting what is known about the sample, how it is constructed and vetted and verified, its limitations, and so on.

4. Collection of minor points:

A. The paper would benefit from a modest revision pass for typographical issues. I observed single-sentence paragraphs and some of the captions had typos. Y-axis labels on the figures would be useful.

B. Figure 4 is difficult to make sense of; it may be possible to improve it using a faceted plot or simply a repeat of the style of Figure 3.

C. I'm not sure it's necessary to publish the QQ plot; a direct visualization of the distribution or a measure of skewness would be reasonable.

D. "technological platforms" (p. 17) seems like an overbroad scope; I think what's meant here is only source control/collaboration platforms like GitHub.

7. PLOS authors have the option to publish the peer review history of their article (what does this mean?). If published, this will include your full peer review and any attached files.

Reviewer #2: No

Reviewer #3: No

---

## [Author Response · Author response to Decision Letter 2]

10 Aug 2021

Our response is included in a separate file.

---

## [Editor Report · Decision Letter 3]

26 Aug 2021

Free and Open Source Software Organizations: A large-scale analysis of code, comments, and commits frequency.

PONE-D-20-37277R3

Dear Dr. Jemielniak,

We’re pleased to inform you that your manuscript has been judged scientifically suitable for publication and will be formally accepted for publication once it meets all outstanding technical requirements.

Kind regards,

Sergi Lozano

Academic Editor

PLOS ONE
---

## [Editor Report · Acceptance letter]

14 Sep 2021

PONE-D-20-37277R3 

Free and Open Source Software Organizations: A large-scale analysis of code, comments, and commits frequency. 

Dear Dr. Jemielniak:

I'm pleased to inform you that your manuscript has been deemed suitable for publication in PLOS ONE. Congratulations! Your manuscript is now with our production department. 

Kind regards, 

on behalf of

Dr. Sergi Lozano 

Academic Editor

PLOS ONE